# High monoclonal neutralization titers reduced breakthrough HIV-1 viral loads in the Antibody Mediated Prevention trials

Daniel B. Reeves [1,2,27] ✉, Bryan T. Mayer [1,27], Allan C. deCamp [1], Yunda Huang [1,2], Bo Zhang [1], Lindsay N. Carpp [1], Craig A. Magaret [1], Michal Juraska [1], Peter B. Gilbert [1,3,4], David C. Montefiori [5], Katharine J. Bar [6], E. Fabian Cardozo-Ojeda [1], Joshua T. Schiffer [1,7], Raabya Rossenkhan[1], Paul Edlefsen [1], Lynn Morris [8,9,10], Nonhlanhla N. Mkhize[8,9], Carolyn Williamson [11], James I. Mullins[2,7,12], Kelly E. Seaton [13,14], Georgia D. Tomaras [13,14], Philip Andrew[15], Nyaradzo Mgodi[16], Julie E. Ledgerwood[17], Myron S. Cohen[18], Lawrence Corey [1,19], Logashvari Naidoo[20], Catherine Orrell[21], Paul A. Goepfert [22], Martin Casapia [23], Magdalena E. Sobieszczyk[24], Shelly T. Karuna [1,25,28] & Srilatha Edupuganti[26,28]

The Antibody Mediated Prevention (AMP) trials (NCT02716675 and NCT02568215) demonstrated that passive administration of the broadly neutralizing monoclonal antibody VRC01 could prevent some HIV-1 acquisition events. Here, we use mathematical modeling in a post hoc analysis to demonstrate that VRC01 influenced viral loads in AMP participants who acquired HIV. Instantaneous inhibitory potential (IIP), which integrates VRC01 serum concentration and VRC01 sensitivity of acquired viruses in terms of both IC50 and IC80, follows a dose-response relationship with first positive viral load ($p = 0.03$), which is particularly strong above a threshold of IIP = 1.6 ($r = -0.6$, $p = 2e\text{-}4$). Mathematical modeling reveals that VRC01 activity predicted from in vitro IC80s and serum VRC01 concentrations overestimates in vivo neutralization by 600-fold (95% CI: 300–1200). The trained model projects that even if future therapeutic HIV trials of combination monoclonal antibodies do not always prevent acquisition, reductions in viremia and reservoir size could be expected.

For those not living with HIV-1, pre-exposure prophylaxis with ART (PrEP) is safe and highly effective in preventing HIV-1 acquisition, though comprehensive implementation remains challenging due to limited availability, uptake, and adherence in heavily affected communities[1–8].

Ultimately, a safe and efficacious preventative HIV-1 vaccine may be required to eradicate HIV/AIDS[9]. Unfortunately, of 9 preventative HIV-1 vaccines tested in efficacy trials, none have achieved strong success[10–18]. Passive immunization with broadly neutralizing antibodies (bnAbs) is a promising strategy to expand the HIV-1 prevention toolbox[19,20], has been extensively documented to prevent mucosal transmission in non-human primates (NHP)[20,21], and may inform future vaccines that elicit bnAbs[22].

The Antibody Mediated Prevention (AMP) trials were the first human efficacy studies of infused bnAbs for HIV-1 prevention (HVTN 704/HPTN 085 and HVTN 703/HPTN 081)[23]. The efficacy of a CD4-

binding site bnAb, VRC01, was tested in men, women and transgender individuals who were vulnerable to HIV-1 in the Americas, sub-Saharan Africa and Europe[24–26]. A total of 4623 participants were enrolled and randomized 1:1:1 into 3 arms, receiving 10 or 30 mg/kg of VRC01 or saline placebo by intravenous infusion every 8 weeks (10 total infusions). Across arms, 174 participants were diagnosed with HIV-1 infection[23]. In the pooled VRC01 groups vs. placebo, prevention efficacy in HVTN 704/HPTN 085 was 27% ($P = 0.15$; 95% confidence interval [CI]: −12, 52%) and in HVTN 703/HPTN 081 was 9% ($P = 0.7$; 95% CI: −45, 43%)[23]. The dose levels were designed based on data suggesting most circulating HIV-1 viruses were relatively sensitive to VRC01 (80% inhibitory concentrations, or IC80 < 10 μg/ml). However, 49% of observed acquired isolates had IC80 > 10 μg/ml. Secondary pre-specified analyses showed that estimated prevention efficacy decreased for higher IC80s (i.e., more VRC01-resistant isolates), with prevention efficacy estimates >80% for HIV-1 viruses with IC80 < 0.3 μg/ml and steadily decreasing to near zero for HIV-1s with IC80 > 5 μg/ml[23].

The predicted serum neutralization 80% inhibitory dilution titer (PT80) biomarker expresses the neutralizing ability of a certain concentration of bnAb against a given virus[27,28]. It is calculated by dividing the bnAb serum concentration at a given time by the IC80 of the bnAb against that virus (also termed inhibitory quotient[29]). This unitless quantity conveniently encapsulates the full heterogeneity of potency from both trial concentrations (range 0–30 μg/mL) and IC80s (observed range 0.1 to 100 μg/ml). Importantly, PT80 was shown to associate with VRC01 prevention efficacy in the AMP trials[27], adding evidence to this biomarker as a surrogate endpoint for HIV-1 acquisition. 90% prevention efficacy was projected for future trials achieving average PT80s > 200, or concentrations 200-fold above the in vitro IC80s of exposing viruses for the duration of follow-up. Here, we also considered an additional metric, the instantaneous inhibitory potential (IIP) that was previously used to quantify ART[30] and bnAb potency. It uses both IC50 and IC80 data to encode how rapidly (and nonlinearly) neutralization rises with titer and provides a scale that clearly distinguishes high e.g., 90% neutralization (IIP = 1) from extremely high neutralization e.g., 99.9% (IIP = 3).

Viral load has long been associated with HIV-1 pathogenesis and progression[31]. Early moments of HIV-1 infection can dictate reservoir creation and immune response[32–34]. Therefore, reductions in viral load during acquisition could improve HIV-1 prognosis[35]. Previously, breakthrough infections among some PrEP users appear to admit lower viral loads[36].

Here, we investigate whether and how VRC01 modulated HIV-1 kinetics in VRC01 recipients who acquired HIV-1 during the AMP trials. We combined pharmacokinetics to predict serum levels of VRC01 and in vitro pharmacodynamics to estimate the time-varying titer of VRC01 against each participant's acquired isolate. Then, we integrated neutralization – a mechanistic extension of the published protection titer correlate[27] – into a mathematical model of viral load and estimated the instantaneous effect of VRC01 on HIV-1 viral load over time. Our analysis found that suppressing viral load in vivo requires much higher levels of VRC01 than would be predicted by in vitro experiments, illuminates likely fitness effects for resistant HIV-1 isolates, unveils how suppressing viremia may be more difficult than preventing infection with bnAbs, and informs dose-selection for future trials of multi-drug combinations.

## Results

### Comparing viral loads by treatment and VRC01 sensitivity of acquired viruses

To assess post-acquisition viral loads, we analyzed 608 viral load observations without ART from the 162 of 174 AMP participants who acquired HIV whose acquired viruses were tested for VRC01 neutralization sensitivity by the TZM-bl assay[37–39]. Approximately 70% of participants had 3 or more viral loads preceding ART initiation, with 30% having more than 4 (Supplementary Table 1).

Across trials (HVTN 704/HPTN 085 and HVTN 703/HPTN 081), there were no obvious differences in viral load trajectories by treatment arm (Fig. 1a). Moreover, there was a wide range in VRC01 sensitivity (IC80) of acquired viruses across participants in both placebo and pooled treatment arms (Fig. 1b). Based on these observations, we combined the 10 and 30 mg/kg VRC01 dose participants into a VRC01 pooled group and used a pre-specified threshold[23] for acquired virus sensitivity to VRC01 -- sensitive: IC80 < 1 μg/ml and resistant: IC80 ≥ 1 μg/ml -- to define four treatment/sensitivity groups for the proceeding analyses: placebo sensitive ($N = 17$), placebo resistant ($N = 45$), VRC01 pooled sensitive ($N = 9$) and VRC01 pooled resistant ($N = 87$).

### Viral loads are transiently reduced in VRC01 recipients who acquired sensitive viruses

Previously, passive administration of VRC01 transiently reduced viremia in PWH not on ART[40]. Here, viral loads in the VRC01 pooled sensitive group (lower left panel, Fig. 1c) appeared the most heterogeneous and potentially lowest, particularly at early time points. To formally test differences between groups, we compared first positive, early (all viral loads within 3 weeks but after first positive), and later viral loads (Fig. 1d); sampling sparsity made common virologic metrics like peak and set point difficult to define for each participant. First positive and early viral loads were lowest in the VRC01 pooled sensitive group ($p < 0.005$ when compared to placebo sensitive, $p < 0.05$ when compared to other groups at first positive, and $p < 0.004$ when compared to other groups at early time points). However, after 3 weeks post first positive, viral loads were not significantly different in any group (Fig. 1d).

### Trial geographic region, sex assigned at birth, and acquired isolate HIV-1 subtype are less predictive of first positive viral load than treatment/sensitivity group

Viral loads may naturally differ by sex, acquired HIV subtype (clade), and/or geographic region[41,42]. To rule out these potentially confounding factors, we examined first positive viral loads by trial, geographic region, and viral subtypes and found no differences (H-test $p = 0.3$, Fig. 1e). However, these variables were highly overlapping: viral subtypes were sequestered by geographic region, and the two studies enrolled female and male participants (sex assigned at birth). Specifically, in HVTN 703/HPTN 081, almost all the isolates were subtype C and participants were female, whereas in HVTN 704/HPTN 085, the isolates were a mix of A, B, D, F, and recombinants of these subtypes and participants were male. As a result, we were unable to compare viral loads across sex, subtypes, and/or geographic region independently.

### Multiple viruses isolated from participants had similar VRC01 sensitivity

As a proxy for within-host diversity, which could influence viral dynamics, and potentially even relate to bnAb-mediated viral escape, we compared IC50 and IC80 values across viruses that were isolated from the same participants ($N = 64$ participants had more than 1 isolate). Viral sensitivities to VRC01 were highly correlated within individuals (Spearman rho = 0.7, Supplementary Fig. 1), suggesting that the within host-quasispecies did not encounter strong post-acquisition selection pressure by VRC01. On a practical level, this finding supports using a single IC80 for each participant. Henceforth, we use the conservative option, the least sensitive isolate (i.e., with the largest IC80) to define the IC80 for each participant.

### First positive viral load reduction via direct (VRC01) and indirect (e.g., fitness cost) effects

Although we did not see statistically significant different first positive viral loads across protocols/regions/clades, we sought to precisely quantify differences between groups and adjust for any minor confounding. Therefore, we performed a regression model to predict first positive viral load by group, adjusting for protocol and region

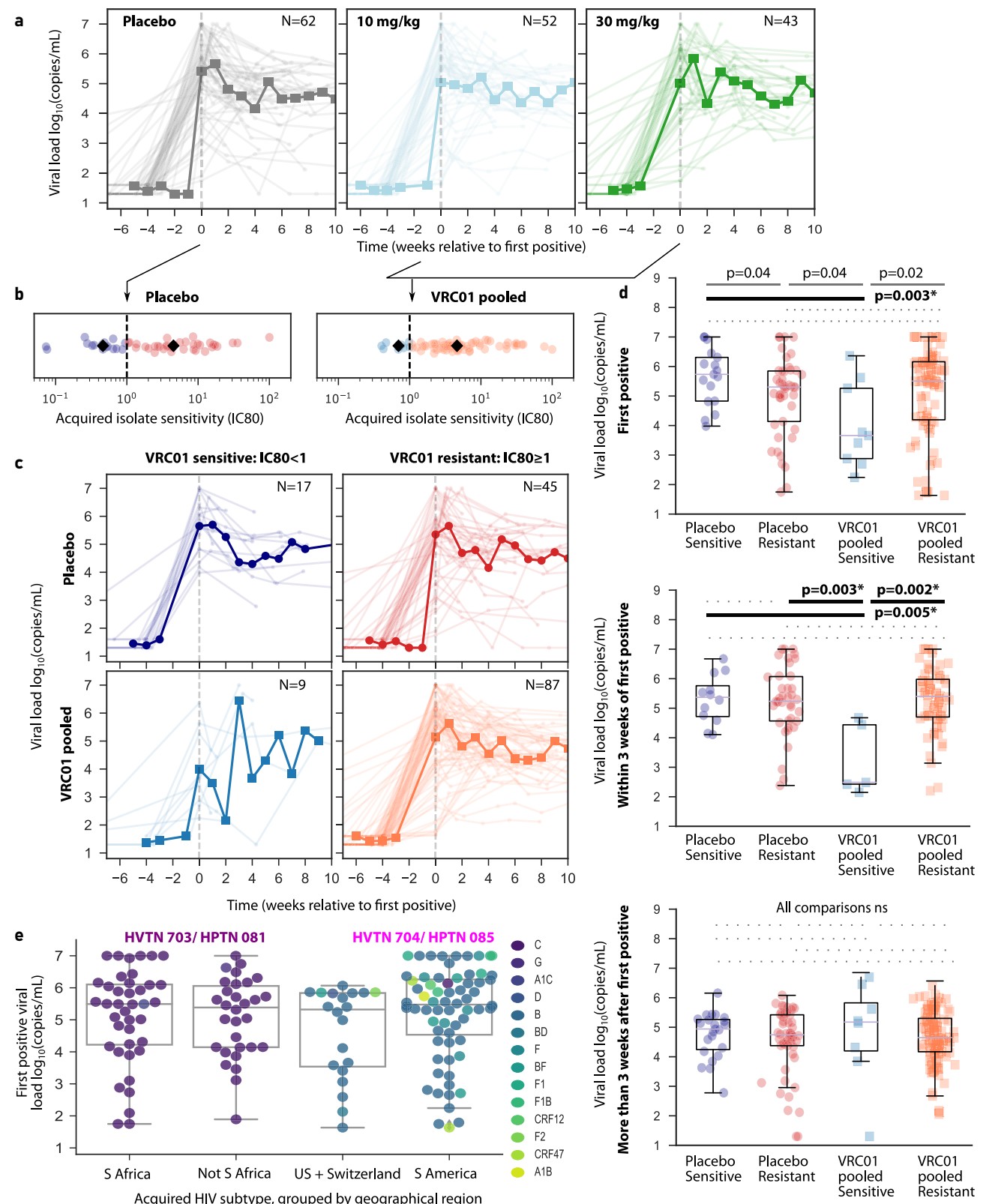

(Methods). From this analysis, the placebo-sensitive group had the highest mean first positive viral loads. The placebo and treated resistant groups were both roughly 0.5 logs lower. We hypothesize this could reflect an indirect effect relating to reduced fitness of highly resistant variants. Finally, the VRC01 pooled sensitive group was a further 1.1 logs (95% CI: 0.1, 2.1, $p = 0.03$) lower than those of the VRC01-pooled resistant group and 1.6 logs (95% CI: 0.4, 2.7, $p = 0.007$)

lower than the placebo sensitive group (Fig. 2). We term this latter reduction the direct effect of VRC01 on viral load.

## No observable delays in ART initiation by trial, treatment arm, or VRC01 sensitivity

To briefly assess clinical implications related to viral load differences, we used cumulative incidence curves to make between-group

**Fig. 1 | Outcomes and longitudinal viral loads from participants who acquired HIV-1 during the AMP studies. a** Pre-ART viral loads pooled across trials and stratified by treatment arm -- placebo (gray; $N = 62$), 10 mg/kg VRC01 (light blue; $N = 52$), and 30 mg/kg VRC01 (green; $N = 43$). **b** VRC01 sensitivity of the acquired isolate for the placebo and VRC01 pooled treatment arms. Dashed vertical line indicates resistant and sensitive threshold for viruses, defined as IC80 ≥ 1 μg/ml and IC80 < 1 μg/ml, respectively. Black diamonds indicate group median for each of four treatment/sensitivity groups: placebo sensitive (navy; $N = 17$), placebo resistant (red; $N = 45$), VRC01 pooled sensitive (blue; $N = 9$), and VRC01 pooled resistant (orange; $N = 87$). **c** Pre-ART viral loads by treatment/sensitivity groups. In **a**, **d**, time values are presented relative to first positive viral loads such that negative days indicate the last negative viral load visit. Thick lines are geometric means binned by week. **d** Viral loads shown by treatment arm and VRC01 sensitive/resistant status grouped into first positive (placebo sensitive: navy, $N = 17$ placebo resistant: red, $N = 45$; VRC01 pooled sensitive: blue, $N = 9$; VRC01 pooled resistant: orange, $N = 87$),

values after first but before 3 weeks (placebo sensitive: navy, $N = 12$; placebo resistant: red, $N = 39$; VRC01 pooled sensitive: blue, $N = 5$; VRC01 pooled resistant: orange, $N = 64$), and values after 3 weeks (placebo sensitive: navy, $N = 23$; placebo resistant: red, $N = 53$; VRC01 pooled sensitive: blue, $N = 8$; VRC01 pooled resistant: orange, $N = 108$). Filled circles indicate the placebo arm, and squares indicate VRC01 pooled. Dashed lines indicate non-significant comparisons, solid lines indicate $p < 0.05$, and bold lines indicate statistical significance after correcting for the 6 comparisons $p < 0.008$ (1-sided Mann–Whitney U test). **e** First positive viral loads grouped by geographic region -- South Africa ($N = 41$), Not South Africa ($N = 31$), US+Switzerland ($N = 22$), and South America ($N = 68$) -- and colored by HIV-1 subtype. Population distributions are not significantly different, $p = 0.3$ by Kruskal-Wallis H-test indicates no differences between groups so individual comparisons were not made. In all plots, box plots indicate median, IQR (box) and 1.5x IQR (fliers).

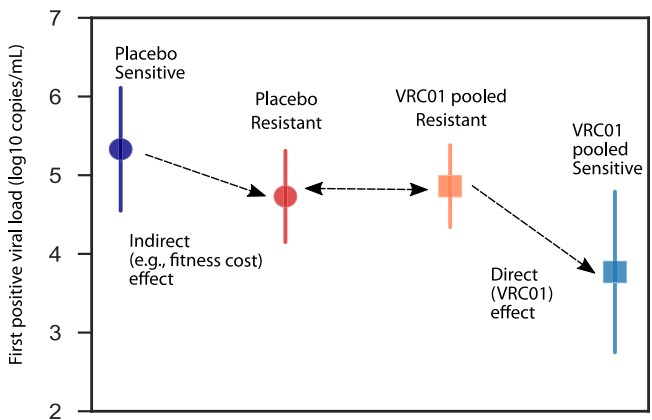

**Fig. 2 | Summary of direct (VRC01) and indirect reductions to first positive viral load.** Mean (dot) and 95% confidence interval (line) first positive viral loads for the following groups: placebo sensitive (navy; $N = 17$), placebo resistant (red; $N = 45$), VRC01 pooled sensitive (blue; $N = 9$), and VRC01 pooled resistant (orange; $N = 87$), calculated adjusting for protocol and study geographic location using a regression model (Methods).

comparisons of the length of time between the last negative visit and the first positive visit as well as the length of time between the first positive visit and ART initiation. No significant differences were detected between trials, treatment arms, and treatment/sensitivity groups (all log-rank test two-sided $p$-values > 0.1, Supplementary Fig. 2). These findings suggest that viral load detection was not heavily delayed by VRC01 direct effects. Similar fractions of participants initiated ART across treatment/sensitivity groups (Supplementary Table 2), and this analysis indicates ART was not initiated earlier or later in any specific group despite observed differences in viral loads.

**Combined PKPD activity as dose-response relationships reveal increasing VRC01 activity lowered first positive viral loads**
We hypothesized that direct effects could follow a dose-response such that higher VRC01 levels result in lower viral loads. To test this dose response, we needed to project VRC01 concentrations at commensal times of viral load measurements. Therefore, we used a pharmacokinetic (PK) model built on the AMP data[27] in which VRC01 infusions follow participant visits and VRC01 is assumed to enter the serum compartment, circulate peripherally, and clear without any influence of HIV viral load (Fig. 3a, Methods). We then projected VRC01 concentrations for each participant and, using their acquired virus IC80, calculated the PT80 at the time of their first positive viral load

measurement using their acquired virus IC80 (example in Fig. 3b, see all participants in Supplementary Data 1).

By predicting titers at first positive, we tested our hypothesized continuous relationship between instantaneous VRC01 activity and viremia. VRC01 concentration at first positive did not significantly associate with first positive viral loads (Fig. 3c, $p = 0.7$). A weak positive correlation was observed between acquired virus sensitivity and first positive viral load in the VRC01 pooled group (Fig. 3d, $r = 0.2$, $p = 0.07$).

However, we also tested more generalized combined PKPD activity-responses, whereby concentration and IC80 were simultaneously used to define relationships between titer (PT80) and viral load. Higher log10 PT80 weakly associated with lower first positive viral loads (Fig. 3e, $r = -0.18$, $p = 0.08$).

Finally, using the metric IIP like PT80 as a combined PKPD activity variable, we observed a significant, but nonetheless relatively weak, relationship between VRC01 activity and viral load (Fig. 3f, $r = -0.23$, $p = 0.03$). This metric appears particularly useful for VRC01 because it uses both IC50 and IC80 and we observed nonlinear increases in neutralization between IC50 and IC80 (Hill coefficient, which encodes nonlinearity of neutralization vs titer, had a mean $h = 1.24$, Supplementary Fig. 3).

**Association between combined PKPD activity and first positive viral load is stronger above a VRC01 activity threshold**
The results of PKPD dose-response analysis led us to question whether VRC01's impact on viral load was effectively negligible below a certain activity level. To examine this latter hypothesis, we applied a segmented linear regression to test whether associations were subject to a threshold effect (Methods). The segmented model outperformed the linear model for both predicted titer (PT80) and IIP (Fig. 3e, f). The segmented model identified change points for certain variables, providing useful thresholds to understand VRC01 effects. Above PT80 = 3 (95% CI: 0.8, 16) the dose response appeared stronger, though significance was limited by data sparsity. For IIPs above IIP = 1.6 (95% CI: 1, 2.2) we documented a robust relationship ($r = -0.6$, $p = 2e-4$) where more VRC01 activity (higher concentration and/or lower IC80) reduced viral loads, subsequently highlighting the utility of this metric.

**Continuous indirect reduction in first positive viral load for resistant infections in placebo participants**
We next questioned if the indirect effect observed for first positives (Fig. 2) between sensitive and resistant acquisitions in placebo participants would exist on a continuum. Indeed, a linear relationship between log10 IC80 of the acquired isolate and first positive viral load was observed in placebo participants (Fig. 3g $r = -0.28$, $p = 0.03$). Importantly, if generalizable, this effect would confound and weaken the observed association between VRC01 and first positive in VRC01 recipients (Fig. 3d–f).

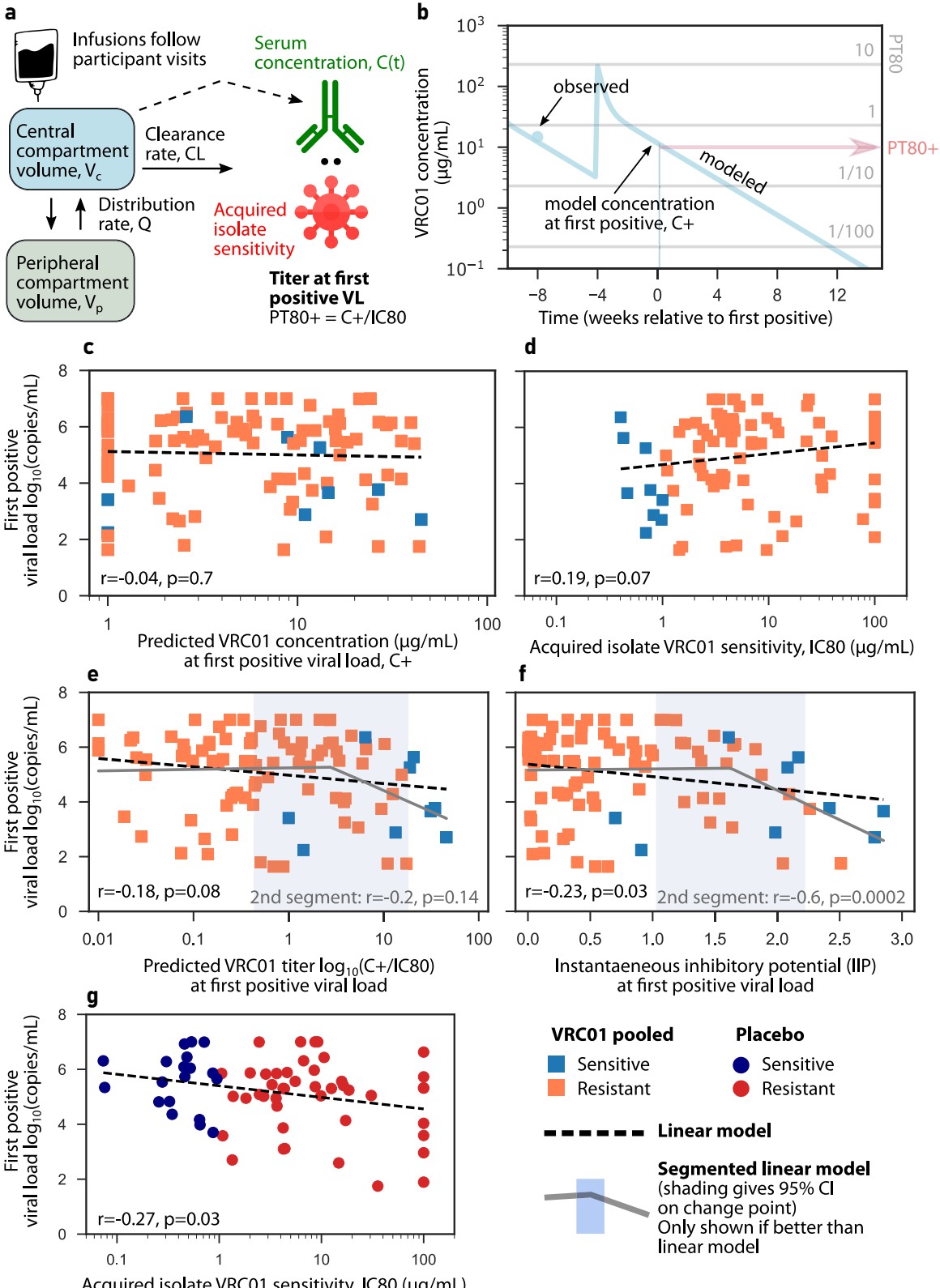

## Virus replication rate indirectly depends on acquired sensitivity to VRC01

To fully describe our longitudinal data and simultaneously estimate indirect and direct effects, we adapted a viral dynamics model originally trained on highly sampled acute HIV-1 viral loads in East African and Thai adults[43,44]. The model encodes virus and cell interactions using a system of differential equations: when virus is introduced into this system, susceptible cells are infected and depleted. Infected cells produce virus and die due to viral cytopathic killing and a mathematical approximation of immune responses [Eq. 4a, Eq. (1)]. We used population nonlinear mixed effects modeling to estimate optimal model parameters[45] for biological rates (Supplementary Table 4).

Our goal was to determine any additional model mechanisms needed to achieve agreement with three observed data types for

**Fig. 3 | Viral load is reduced directly by higher VRC01 activity and/or indirectly, e.g., via VRC01 resistant fitness costs. a** A two-compartment pharmacokinetic (PK) model trained on the present data–model structure was previously designed for VRC01 PK using an AMP case-control cohort (Supplementary Table 4)[88]. **b** Observed (dots) and modeled (line) VRC01 concentrations over time in a representative participant. The horizontal lines illustrate VRC01 concentrations corresponding to PT80 titers of 10, 1, 1/10 and 1/100 against the participant's acquired isolate. **c–g** PK and combined PKPD activity visualized as "dose-response relationships" for variables against first positive viral load. **c** Projected VRC01 concentration at first positive viral load. **d** Acquired virus sensitivity to VRC01 in VRC01 pooled group. **e** Predicted 80% titer (PT80) at first positive. **f** Instantaneous inhibitory potential (IIP). **g** Acquired virus sensitivity to VRC01 in placebo group. Squares and blue/orange colors indicate sensitive/resistant VRC01 pooled participant data, whereas dots and navy/red colors indicate sensitive/resistant placebo participant data. Black dashed line is a linear regression line (Pearson correlation coefficient $r$ and $p$-value noted in panel title). In **e**, **f** two-segment dose-response models described data better than the linear model. In those cases, the segmented model is shown as a solid gray line with a blue shaded 95% confidence interval around the change point of the segments.

each participant simultaneously: in vitro IC80 of acquired virus, longitudinal VRC01 concentrations, and longitudinal viral loads. Therefore, we sequentially added mechanisms to the model and tested them against AMP data, selecting the optimal model that minimized the corrected Bayesian Information Criterion (BICc)[46], a metric that balances accuracy of model fit against model complexity.

First, model agreement to placebo participant data was optimized by including indirect effects expressed as a viral production rate $\pi(IC80)$ that depended on IC80 (Supplementary Table 5; note fit was statistically similar but slightly worse using IC80-adjusted virus infectivity rate, $\beta$). We then fit the model to VRC01-recipient data including the indirect effects and estimated the direct effect at the same time. Importantly, agreement to these data could not be achieved by directly imputing VRC01 concentration and IC80 (model BICc was over 3000 points worse than the optimal model that we eventually selected, Supplementary Table 6).

### VRC01 activity in vivo is significantly overestimated by serum VRC01 concentrations and in vitro IC80s
Based on published data[23,27,28,47,48], we hypothesized that in vitro values of IC80 overestimate in vivo neutralization potential. Therefore, for the direct effect we defined a potency reduction factor $\rho$, which scales between in vitro and in vivo titer. As an example, at PT80 = 1, the serum concentration equals the IC80 and 80% of viruses would be expected to be neutralized in the TZM-bl assay. With $\rho$=100, the same titer achieves less than 10% neutralization, and we require PT80 = 100 to achieve 80% in vivo neutralization. Although we could not estimate a person-by-person value, the model's ability to recapitulate data was substantially improved by including a potency reduction factor with a population average of $\rho$=628 (95% CI: 313, 1262) (trajectories in Fig. 4b & model fitting score vs values of $\rho$ in Supplementary Fig. 4). This suggests in vitro titers overestimate effective in vivo titers by approximately 600-fold.

In comparison, the mean VRC01 PT80 titer required to achieve 80% prevention efficacy was roughly 50 and 120 for non-human primates and humans, respectively[27]. This suggests a ~100-fold overestimate between in vitro titer and in vivo activity for prevention efficacy, lower than our potency reduction factor by roughly 6-fold. The difference could indicate that prevention requires lower VRC01 titers than those needed to suppress viremia during early systemic infection.

### Optimal model of VRC01 viral kinetics
Once the potency reduction factor was included, our model projected smoothed VRC01 and viral load trajectories that quantitatively match all observed data (8 example fits are shown in Fig. 4b and all participant fits are provided in Supplementary Data 1 & 2). For individuals with high VRC01 titers (concentrations several logs above IC80), including some who received infusions after first positive, viral loads are instantaneously modified – see model fits in rightmost 3 panels of Fig. 4b. Note, VRC01 infusions were not knowingly given after an individual had acquired HIV. However, 93 participants, approximately evenly distributed across trial arms, were retroactively determined to

have had viral loads when they received an infusion (Supplementary Table 3).

To further illustrate why potency reduction was required, we show estimated model projections in the absence of potency reduction (dashed lines in Fig. 4b). Unsurprisingly, viral load trajectories are unrealistic as the model imposes nearly complete neutralization when concentrations are several logs above IC80s, in those cases, it is difficult to reconcile the observed viral loads. Additionally, without potency reduction, model parameter estimates are unstable, in particular, viral infectivity is estimated to be unrealistically high to account for the observed viral loads despite high titers of VRC01 (Supplementary Fig. 5).

### Viral dynamics model-based acquisition timing agrees with other approaches
A key parameter estimated by the mechanistic model is acquisition time[49]. We used this value and the PK model to estimate PT80 at the estimated date of acquisition. PT80 estimates at the time of acquisition from our PKPD-V model resembled those of a published model that relies on clinical and sequence diversification data[50] (Fig. 5a). Additionally, both models predicted no observed acquisitions with PT80 > 100, levels that are theoretically possible given observed concentrations and IC80s. This finding is consistent with the prevention efficacy previously reported[23], reflecting that some acquisition events may have been averted due to therapeutic VRC01 levels.

### Model-interpolated viral load metrics indicate some differences between study groups
Based on the sparse sampling, raw data estimates of common metrics such as peak, area under curve (AUC), and/or setpoint viral load were not well defined. Therefore, we projected these metrics using the model (Fig. 5b). We observed a trend that model-estimated peak viral loads were the lowest among VRC01 recipients who acquired VRC01-resistant isolates compared to other groups, suggesting indirect effects had a greater impact on peak than some other metrics. Then, using the regression model (Methods), AUC during 3 months after first positive was found to be 0.5 logs lower in the VRC01-pooled sensitive group compared to both the placebo sensitive and the VRC01-pooled resistant groups ($p$ = 0.04 for both comparisons, Fig. 5b). Regression models found set points were not different across groups, consistent with waning VRC01 concentrations and therefore waning direct effects by the time of set point.

### In simulation, higher titer therapies could block acquisition and/or substantially blunt viral loads and reduce reservoir creation
An important rationale for developing and training the mechanistic model was the ability to plausibly simulate future scenarios with increased PKPD activity -- i.e., higher titer through better PK and/or more potent single bnAbs and combination bnAbs. Recently, Mdluli et al. also showed that viral load area under the curve (AUC) during primary infection is predictive of time to viral rebound after ART suppression and analytical treatment interruption[51]. Therefore, we performed a simulation study projecting longitudinal viral loads and AUC (as a surrogate for "reservoir size") under different bnAb activity.

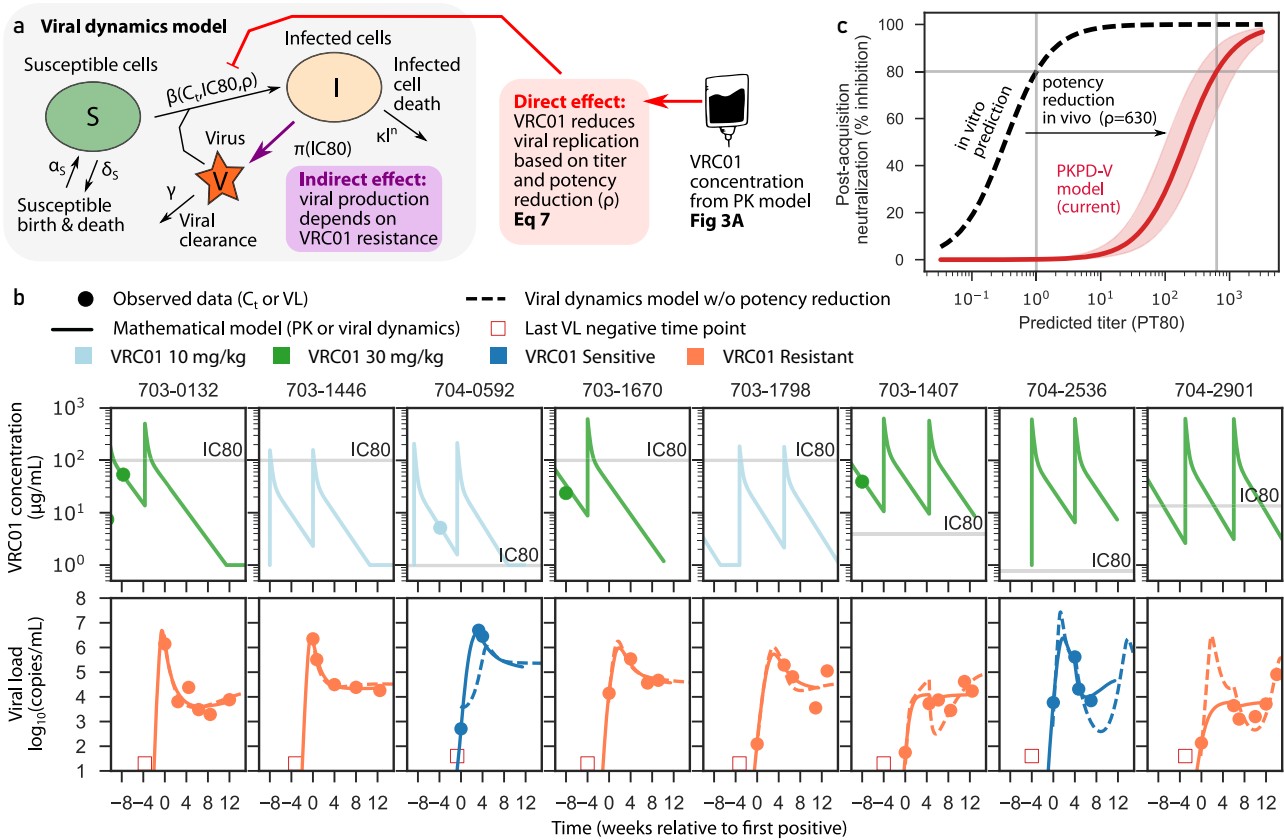

**Fig. 4 | The optimal model for AMP viral loads incorporates VRC01 concentration and acquired isolate sensitivity to VRC01, as well as fitness costs and in vivo potency reduction. a** The viral dynamics and PKPD (PKPD-V) mathematical model used to impute viral load curves[43]. Susceptible cells S are born and die in the absence of virus (rates $\alpha_S$ and $\delta_S$, respectively) and are infected upon exposure to virus V with infectivity rate $\beta$, infected cells I die nonlinearly with rate $\delta_I(I) = \kappa I^n$, virions are produced with rate $\pi$ and cleared with rate $\gamma$. VRC01 concentration is predicted by the same PK model as in Fig. 2a and is connected into the viral dynamics through the PD model where VRC01 concentration reduces cell infection events, $\beta(C_t, IC80, \rho)$ based on IC80 of acquired virus and in vivo potency reduction. The indirect effect implies viral production is lowered for more resistant isolates $\pi(IC80)$. **b** Eight examples of participants with simultaneous PK and viral load fits relative to first positive viral load (see all in Supplementary Data 1 & 2). Dots represent observed concentrations and viral loads, solid lines represent model output, and dashed lines represent counterfactual model simulations with no potency reduction. PK curve colors indicate treatment arm (light blue: 10 mg/kg; green: 30 mg/kg). Viral load curve colors indicate acquired virus sensitivity to VRC01 (blue: VRC01 sensitive; orange: VRC01 resistant). Annotations provide definitions for model-based metrics. Horizontal gray lines in PK plots contextualize serum concentrations relative to in vitro IC80s. **c** Predicted % of cellular infections blocked at a given titer estimated from in vivo viral load data (solid red) compared to in vitro predicted neutralization (dashed black). The pink shading around the solid red curve denotes 95% confidence intervals. The difference between the curves illustrates the 600-fold potency reduction in vivo.

In simulations with higher titer antibodies, we project kinetics would be noticeably different in the sensitive group for a theoretical regimen with 10x (or more) the VRC01 titer compared to AMP (Fig. 5c). If a combination bnAb regimen achieved at least 100x higher titer than AMP, we project viral loads in both sensitive and resistant groups would look markedly different from natural infection (no VRC01) (Fig. 5c). As a result, viral load AUC is modulated and therefore we predict reservoir sizes would be substantially lower (Fig. 5d). In this >100x titer range, our model also begins to predict post exposure prophylaxis in some cases (see flat lines at limit of detection 50 copies/mL in both sensitive and resistant groups). Quantitatively, we predict peak viral loads and reservoir sizes would be a median of 3-logs lower for sensitive acquisitions with 10x titer, whereas this same drop would require between 100–1000x titer for resistant acquisitions (Fig. 5e). Together, if future antibody regimens can reach PT80 > 100 against all viruses in an individual's quasispecies, both overall protective efficacy and post-acquisition modulations of viral loads that reduce HIV reservoirs should be expected.

## Discussion

The AMP studies provided proof-of-concept that HIV-1 acquisition can be prevented by a broadly neutralizing antibody in humans. Here, we studied participants who acquired HIV-1 in the AMP studies and carefully characterized the viral loads of placebo and VRC01 recipients using mathematical models.

We identified a direct effect of VRC01 on post-acquisition viral loads which followed a "dose-response" relationship between combined VRC01 PKPD activity and first positive viral load. The effect was particularly noticeable using instantaneous inhibitory potential (IIP), which incorporates VRC01 concentration and the IC50 and IC80 of an acquired virus into a single metric that accounts for nonlinearity in neutralization. Moreover, a threshold was found (IIP = 1.6) above which the relationship between IIP and viral load was much stronger ($r = -0.6$).

However, the direct effect was transient, likely because of waning VRC01 concentrations, and potentially even an effect whereby viral binding saturates antibodies and further reduces VRC01 levels. Nevertheless, modeled viral load area under the curve was lower in the treated sensitive group. Blunting or delaying HIV-1 viremia might be relevant for pathogenesis[52], reservoir creation[51], viral evolution[53], and/or onward transmission during acute infection. Furthermore, small but exciting studies[54,55], showed other bnAbs can achieve prolonged (up to 2 years) HIV-1 suppression after stopping ART potentially due to a vaccinal effect in which infused bnAbs interact with virus, e.g., forming immune

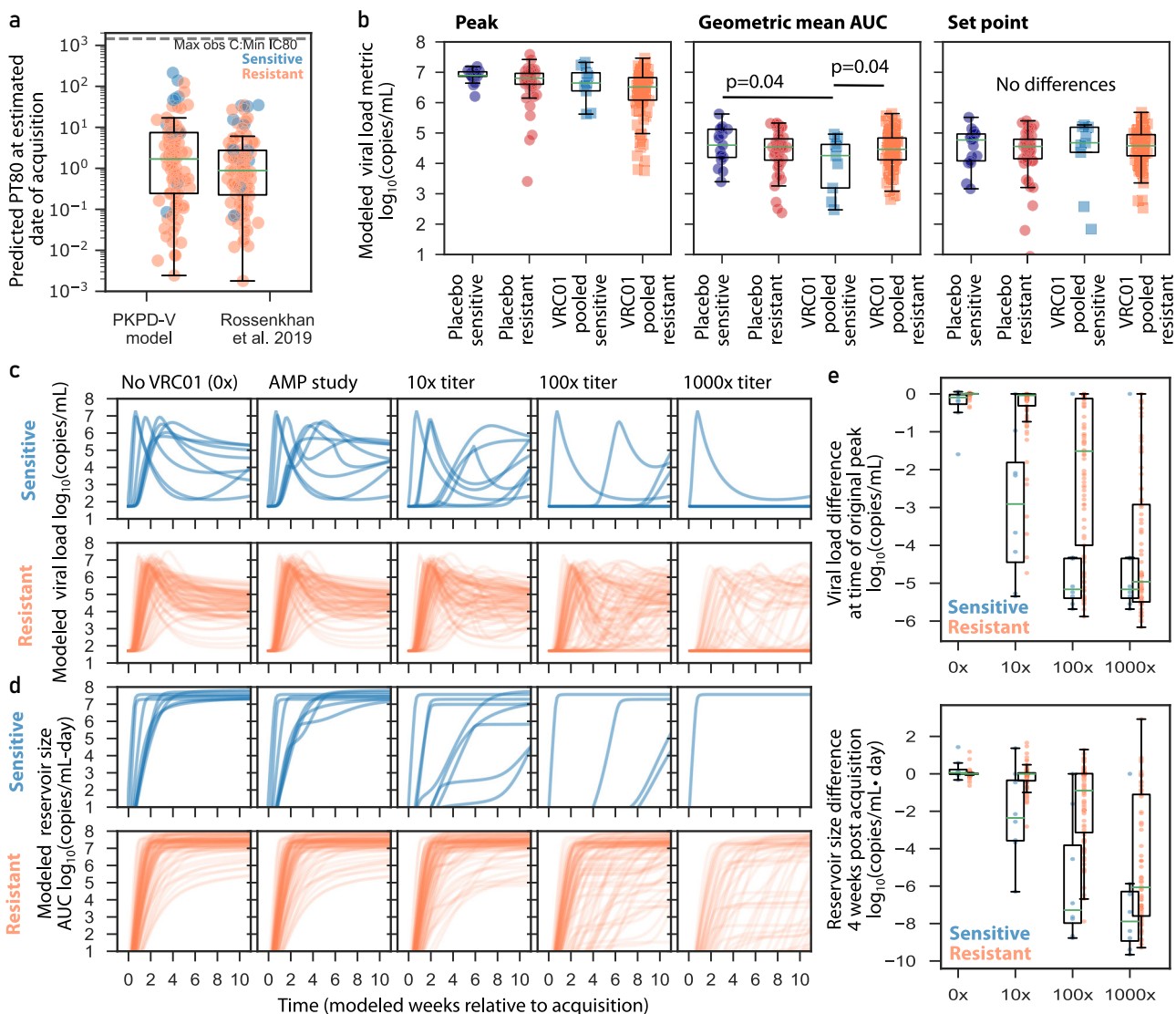

**Fig. 5 | Modeled acquisition time, viral load metrics, and simulations of viral load and reservoir size during future high-titer trials. a** Predicted titer (PT80) at estimated date of infection using our model compared to a published infection timing approach[50]; orange and blue dots indicate resistant (N = 84) and sensitive (N = 11) isolates, respectively, from VRC01 recipients with available timing estimates. The maximum measured concentration divided by the minimum observed IC80 is shown as a dashed horizontal line to contextualize a theoretical upper bound on PT80s. **b** Model estimated metrics for the following treatment and resistance groupings: placebo sensitive (navy; N = 17), placebo resistant (red; N = 45), VRC01 pooled sensitive (blue; N = 9), and VRC01 pooled resistant (orange; N = 87). Each dot represents a participant model value and box plots indicate median, IQR (box) and 1.5x IQR (fliers) across participants. The *p*-values are two-sided and unadjusted for multiple comparisons, and were obtained from a linear regression model adjusting for protocol and study geographic location. Model peaks were not statistically compared. **c** Modeled longitudinal viral loads from all VRC01 recipients split by sensitive (blue lines) and resistant (orange lines) viruses. **d** Modeled longitudinal HIV "reservoir size" (assumed proportional to viral load

area under the curve) from all VRC01 recipients split by sensitive (blue lines) and resistant (orange lines) viruses. In **c**, **d** from left to right panels indicate counterfactual of participants without VRC01, AMP participants as in the actual trial, and 3 different levels (10x, 100x, and 1000x) of higher-potency trials. There are fewer lines in some plots because the modeled higher potency scenarios sometimes completely suppress viremia below limit of detection. **e** Differences between AMP study and other modeled scenarios. Viral load difference is calculated by subtracting off (on log10 scale) the peak AMP study viral load from each theoretical scenario in **c** where viral load is calculated at the original time of peak relative to acquisition. Reservoir size difference is calculated by subtracting off the week 4 AMP study viral load from each theoretical scenario in **c** where viral load is also calculated at week 4 relative to acquisition. Each dot indicates a participant's difference (on log10 scale) from each new scenario relative to their baseline (AMP study) value. Box plots indicate median, IQR (box) and 1.5x IQR (whiskers), calculated from VRC01 pooled sensitive (blue; N = 9), and VRC01 pooled resistant (orange; N = 87). In all panels, box plots indicate median, IQR (box) and 1.5x IQR (fliers).

complexes, to enhance immune control[56]. We did not see signs of a vaccinal effect for VRC01 here, which may be related to the short duration of the study, properties of VRC01, or the overwhelmingly high viral loads and/or unprimed immune system during primary infection. Indeed, in certain instances vaccinal effects only manifested after longer times[57], and because of the absence of any signals we did not apply the more complicated models that could predict when control might emerge[58,59].

We also observed indirect effects where VRC01 resistance predicted lower first positive viral loads in the placebo group. This effect could indicate fitness costs related to resistance to VRC01 -- in our optimized mathematical model, data fit was improved when virus production decreased with IC80. As VRC01 binds to the CD4 binding site, which is necessary for viral entry[60], resistant viruses might also lose some ability to enter cells. Indeed, viruses that are highly resistant to VRC01 have been shown to have compensatory

fitness costs[40,61]. Future studies are warranted to assess fitness directly.

The mathematical model was capable of fitting to longitudinal VRC01 serum concentrations and viral loads simultaneously by adding a VRC01-direct neutralization effect, while integrating the sensitivity of each participant's acquired virus against VRC01. The model fit the data best when both direct and indirect effects were included. Importantly, modeling allowed simultaneous estimation of these two effects. Finally, we demonstrated simulations of counterfactual and future scenarios using the model, paving the way for usage in trial design.

A key observation was that the model could not fit to all data without including a factor that accounts for in vivo potency reduction. For VRC01, this phenomenon has qualitatively been observed in human studies outside of primary infection: HIV-1 could be suppressed by VRC01 without ART but viral loads often restabilized before VRC01 levels decayed below the IC80[40], sometimes (but not always) because of the emergence of resistant variants[62]. During ART interruption with VRC01 supplementation, viral rebound occurred despite high serum concentrations of VRC01, and the sampled emerging variants were often not obviously resistant[40]. Modeling by Saha et al. required very high estimates of IC80s for rebound variants to reconcile with the VRC01 concentration in an ATI study[63]. Here, we estimated that the in vivo VRC01 PT80 titer required to suppress viremia was 600-fold higher than would be expected from serum concentrations and in vitro IC80 titers, suggestive of in vivo potency reduction.

There are several possible explanations for in vivo potency reduction. The TZM-bl target cell assay used to quantify in vitro neutralization uses cells that are highly permissive to infection[39], potentially allowing for larger reductions by VRC01. Moreover, the pseudoviruses used in the assay may be more VRC01 sensitive than wild-type viruses[64]. It is also possible that VRC01 is not as effective in vivo − affinity and/or potency could be reduced, clumping might occur, and/or anti-idiotype antibodies could reduce some bnAb efficacy. Yet, functional anti-idiotype antibodies were not identified in AMP[65] and serum VRC01 has been shown to maintain neutralization (and some other functions) when re-harvested from participants after infusion/injection[66]. Another explanation may be that cell-to-cell transmission of HIV-1 effectively lowers the ability of VRC01 to find virus to neutralize[67−69]. Perhaps the simplest explanation of all implicates VRC01 biodistribution. Although we sample and model VRC01 concentrations and viral loads in the serum, most viral production occurs elsewhere (e.g., lymph nodes)[70] and viremia is generally assumed to be spillover from these locales. Thus, the potency reduction factor could indicate differences in concentration between serum and those sites. Post-infused levels of VRC01 have been shown to correlate between rectal tissue/secretions and serum[71], but levels were also sometimes 10−100x lower outside of serum. Therefore, additional studies investigating bnAb levels and function in other compartments could be helpful for next-generation regimens.

Viral load blunting and delay was also observed in breakthrough acquisition during ART-mediated prevention trials[36]. Our model illuminates why blunted viral loads and relatively sparse sampling makes the first positive viral load a transient indicator of VRC01 efficacy despite acquisition. Our simulations also demonstrate that blunted viremia meaningfully lowers viral load AUC, which has been linked to HIV reservoir generation[51,72]. Therefore, even if future therapies do not always block acquisition completely, they might potently reduce reservoirs. In contrast to ART monotherapy, which rapidly and deterministically leads to viral escape[73−75], we show some evidence that bnAb monotherapy is relatively robust. Sensitive viruses were observed in participants across all trials and dose arms. More than one isolate was found in ~1/3 of participants and within-host viruses were highly correlated in their sensitivity to VRC01. Thus, we suspect in the time scale of this specific study, no obvious evolution occurred within-hosts such that VRC01 resistant variants did not develop anew. Our

work corroborates other modeling of VRC01 infusion during chronic infection and ATI in which returning and rebounding variants are inferred to have existed at low frequencies vs. emerging de novo[62,63,76].

There are limitations to this analysis. The number of VRC01 recipients who acquired a VRC01-sensitive virus is limited. Viral isolate subtypes were also stratified by geographical location, limiting comparisons by either factor. Comparisons by sex assigned at birth were generally not distinguishable from comparisons across trials due to design and we could not analyze results based on route of transmission, which may alter acquisition probability[77,78]. Our viral load model does not notably consider cell-to-cell transmission, any enhancement of CD8 T cell killing via VRC01, or any active mechanism whereby VRC01 kills infected cells (such as antibody-dependent cellular cytotoxicity or ADCC) and/or mediates phagocytosis[66,79]. This would be particularly important to know given the hypothesized bnAb vaccinal effect mentioned above[56] though it may be less critical here as VRC01 has demonstrated little effector functionality in prior studies[66]. Although we generally expect in vitro inhibitory concentrations to overestimate in vivo efficacy, other bnAbs (and in particular those that do not target the CD4 binding site) may not obey the same potency reduction factor found here[66]. Recent data have now quantified sensitivities of different bnAbs to these viruses[80], paving the way for future analyses. Since VRC01 appears to work predominantly via neutralization, this factor may be an upper limit on this first generation of anti-HIV bnAbs; current bnAbs with enhanced biodistribution or whose activity reflects neutralization plus other functions might have better in vivo potency. If different, it will be important for models of combination bnAbs to include multiple scaling factors to properly predict neutralization in vivo.

The PK model also assumes VRC01 concentration is not modified by viral load levels. Although not studied for VRC01 other bnAbs have had PK assessed in PWH. For PGDM1400, the same 2-compartment model structure was supported, but terminal half-lives may have been 1.7x faster in PWH vs HIV-negative individuals[81]. Given the AMP dosing, this would mean 3-fold lower after 20 days and 10-fold lower after 8 weeks (i.e., an AMP dosing interval, Supplementary Fig. 7). If VRC01 is cleared more rapidly in participants who newly acquire HIV, our present modeling would overestimate VRC01 concentrations, and thus accounting for this difference would admit a lower potency reduction. But, without precise PK data, for now we note that a 3−10 fold reduction in VRC01 concentrations would still be within the confidence intervals we estimate for the potency reduction factor in Fig. 4.

Our model suggests titers above 600 (95% CI roughly 300−1200) are required to reduce viremia by 90%. Yet, prior meta-analysis of non-human primate challenge studies and statistical analyses of AMP found PT80 titers of 90 and 200 (i.e., serum VRC01 concentrations 200 times the in vitro IC80), respectively, would provide approximately 90% protective efficacy[27,82]. It is intriguing that titers required for prevention appear lower than the titers required to reduce viremia. The differences in titer thresholds could be explained by the ability of antibodies to prevent vs control viremia (control might be mediated by T-cell immunity)[83]. Additional variables such as the inoculum size and/or anatomic location, mucosal immunology, and/or viral latency might differ between the early moments of HIV-1 exposure vs systemic infection.

In conclusion, we used data from the most relevant human bnAb prevention studies to train a complete framework and extract mechanistic insight from trial viral load data. We show viral loads are a powerful clinical endpoint in HIV prevention efficacy trials, define neutralization titer and IIP benchmarks for future trials, highlight discrepancies between in vitro and in vivo activity, and release a holistic

system that could be used with reasonable assumptions for the design and optimization of future clinical trials of broadly neutralizing antibodies for HIV-1 prophylaxis and treatment.

## Methods

### Antibody Mediated Prevention (AMP) trials

The present analysis is a post hoc analysis of the HVTN 704/HPTN 085 (NCT02716675) and HVTN 703/HPTN 081 (NCT02568215) randomized controlled trials, for which data on primary and secondary outcome measures were published previously[23]. These trials were designed to test whether passive infusion of the broadly neutralizing antibody VRC01 (vs. placebo) can prevent HIV-1 acquisition. Written informed consent was obtained from all participants. Details on trial design and implementation, characteristics of the enrolled participants, overall efficacy of VRC01, safety and tolerability of VRC01 infusions, and PrEP use have been published elsewhere[23,24,26,65]. All users of PrEP were excluded in the present analysis.

### AMP study participants

As the majority of HIV-1 transmissions occur via the vaginal or rectal mucosa, and bnAb distribution may vary at these sites (with potential implications on prevention efficacy), sex/gender identity was considered in trial design. Therefore, two trials were conducted in two distinct study populations:

HVTN 704/HPTN 085 enrolled 2701 participants who had been assigned male sex at birth or who self-reported as being transgender and who had sex with cisgender men or transgender persons. The numbers of participants by sex at birth were 2675 male, 26 female; by self-reported gender identity: 2443 male, 44 female, 19 transgender male, 136 transgender female, 31 gender queer, 25 gender variant or gender non-conforming, 12 "other", and 13 preferred not to answer[23]. Participants may have reported more than one gender identity. Participants ranged from 18 to 50 years of age and were enrolled at sites in the US, Brazil, Peru, and Switzerland between April 2016 and October 2018[23].

HVTN 703/HPTN 081 enrolled 1924 participants who had been assigned female sex at birth, 18 to 50 years of age, in 7 countries in sub-Saharan Africa between May 2016 and September 2018. Gender identity was not assessed in participants in HVTN 703/HPTN 081 outside of South Africa.

### Post-acquisition viral load analysis

Here, we analyzed the kinetics of previously unpublished longitudinal viral load measurements from participants with confirmed acquisition of HIV-1 infection after enrollment and by the week 80 visit. Only viral load measurements prior to ART initiation were used for modeling/analysis, but the percentages of participants in each treatment arm initiating ART are provided (Supplementary Table 2). Additional information on viral load measurement counts is provided by post-diagnosis study day, infusion timing relative to diagnosis and first positive detection, pre-ART post-acquisition measurement counts by study group relative to diagnosis time, and treatment assignment and IC80 category by days relative to diagnosis (Supplementary Tables 7, 8).

### Inclusion and ethics

All work described here complied with all relevant ethical regulations. The TZM-bl target cell neutralization assay work was approved by the Duke University Health System Institutional Review Board (Duke University) through protocol ID Pro00093087. For the NICD, the TZM-bl target cell neutralization assay work was approved by the University of the Witwatersrand Human Research Ethics Committee through protocol M201105. All participants provided written informed consent. Participants were compensated to cover relevant trial participation costs for each completed study visit.

### TZM-bl target cell neutralization assay

In vitro sensitivity of HIV-1 envelope (Env) pseudotyped viruses to VRC01 was quantified via the TZM-bl target cell assay, performed at Duke University and at the National Institute for Communicable Diseases in Johannesburg, with assay equivalency established between the two labs[37–39]. In brief, Env-pseudotyped viruses were produced by cotransfection of 293T/17 cells (American Type Culture Collection, no. CRL-11268) with env plasmids and an env-defective backbone vector (pSG3delEnv). Eight three-fold serial dilutions (starting dilution: 1:10) of each autologous serum sample were assayed against each HIV-1 Env-pseudotyped virus using TZM-bl target cells (National Institutes of Health AIDS Research and Reference Reagent Program no. ARP-8129). Neutralization titers are the reciprocal serum dilution at which relative luminescence units (RLU) were reduced by either 50% (ID50) or 80% (ID80) relative to virus control wells after subtraction of background RLU in cell control wells. The VRC01 drug product (Leidos Biomedical Research, Inc./VRC-HIVMAB060-00-AB, lot no. 16–524; stock concentrations prepared at Duke and sent to the The National Institute For Communicable Diseases) served as a positive control. Data was collected with the Victor×Light luminometer (PerkinElmer 2030 software, instrument program v.4.00.5) at Duke up to 11 November 2020. After this date, the Glomax Navigator System luminometer was used for data collection using Glomax Navigator software (v.3.2.3, firmware v.4.92.0). At NICD, the PerkinElmer Victor× luminometer was used for data collection with PerkinElmer 2030 software (v.4). See also refs. 23,27.

### Pharmacokinetic modeling to project VRC01 concentration

We used a 2-compartment pharmacokinetic (PK) model (Fig. 3a) to project VRC01 concentrations over time. Parameterization used typical PK notation defining the volumes of the central ($V_c$) and peripheral compartments ($V_p$), as well as the clearance rate ($CL$) and distribution rate ($Q$), respectively. To connect with the classical chemical reaction type parameterization, we can use $CL=k_{out}V_c$, $Q=k_{c:p}V_c$, and $V_p=k_{c:p}V_c/k_{p:c}$, where the rate constants ($k$) indicate transition from compartments and/or the final clearance rate $k_{out}$.

PK model fitting was accomplished using population nonlinear mixed effects modeling in Monolix[45] with a model structure and parameterization determined in previous study of this cohort[27,84]. Using that optimized model (parameters and fitting quality in Supplementary Table 4), individual parameters for each participant were selected as the mode of the individual-level posterior conditional distribution constructed by the MCMC algorithm in the Monolix software. Individual values are also provided in Supplementary Data 3. Using the individual-level parameters, the PK model was used to predict concentrations at the first positive and estimated time of acquisition for each participant. PK trajectories were simulated incorporating repeated VRC01 infusions based on each participant's observed dosing schedule.

### Dose-response curve analysis

Using the SciPy package, we calculated Pearson correlation coefficients between modeled pharmacokinetic (PK) and pharmacodynamic (PD) variables and observed first positive viral loads (Fig. 3). We used the estimated concentration of the VRC01 at the time of first positive $C_+$, the IC80 of the acquired isolate to calculate the PT80 titer at first positive,

$$PT80_+ = C_+ /IC80 \tag{1}$$

which gives the concentration relative to the IC80. Finally, we used instantaneous inhibitory potential,

$$IIP = \log_{10}\left[1 + (C_+ /IC50)^h\right] \tag{2}$$

where $h$ is the Hill exponent from the in vitro neutralization curve. The Hill coefficient was calculated using the IC50 and IC80 as described in ref. 28

$$h = \frac{-\ln(4)}{\ln(\text{IC50}/\text{IC80})} \qquad (3)$$

The Hill slope calculation is undefined in participants whose acquired isolate was experimentally saturated (IC50 = IC80). Because there was no relationship between Hill slope and IC50 (Supplementary Fig. 3), we calculated neutralization (and IIP) for these individuals using the value of IC50 = 100 µg/mL and the population mean Hill slope.

## Segmented linear regression for dose response analysis

We noted visually that certain regions of dose-response curves were much more heterogeneous (noisier) than others, therefore, to assess regions of dose response, we applied a segmented linear regression model with a single change point (implemented in the Python package piecewise_regression[85]). A segmented model (vs. a linear model) was preferred when a Davies test was significant -- this tests for a non-constant regression parameter in a linear model and was deemed significant when $p < 0.05$.

## Population nonlinear mixed-effects model fitting

We also used a population nonlinear mixed-effects (pNLME) approach to model plasma viral loads. Here, observed plasma viral loads are modeled as $\log_{10} V_{ij} = f_V(t_{ij}, \boldsymbol{\theta}_i) + \epsilon_V$ for participant $i$ at time $j$. The function $f_V$ calculates the viral load over time based on the numerical solution to a set of ordinary differential equations [(4) and (5)] parameterized by individual-specific parameter vector $\boldsymbol{\theta}_i$. The residual measurement error was assumed to be log-normal (linear on log10 viral load scale) and parameterized by $\epsilon_V \sim N(0, \sigma_V^2)$. In the pNLME framework, population-level parameters are represented as fixed effects, $\boldsymbol{\theta}_i^{pop}$, with individual-level variation described by random effects, $\eta_i$. The distribution of the random effects was assumed to follow $\boldsymbol{\eta}_i \sim N(0, \boldsymbol{\Omega})$ with $\boldsymbol{\Omega}$ the covariance matrix and the standard deviation of each random effect (diagonals of $\Omega$) was denoted by $\omega_k$ for each parameter, $k$. Specified correlations between parameters $k$ and $l$ (off-diagonals of $\Omega$) were denoted $r_{kl}$. Parameters with log-normal distributions were modeled as $\theta_{ik} = \theta_k^{pop} \exp(\eta_{ik})$. Log10-transformed parameters with normal distributions were modeled as $\log_{10} \theta_{ik} = \log_{10} \theta_k^{pop} + \eta_{ik}$. Parameters without random effects have no variation between individuals ($\omega_k = \eta_{ik} = 0$).

## Mechanistic models for viral load and indirect effects

We mechanistically modeled viral load using extensions of a system of ordinary differential equations successfully used previously to model HIV-1 primary infection viral loads in a natural history study[43] (over-dot denotes derivative in time):

$$\dot{S} = \alpha_S - \delta_S S - \beta S V$$

$$\dot{I} = \beta S V - \kappa I^n I \qquad (4)$$

$$\dot{V} = \pi I - \gamma V,$$

Model optimization was performed via maximum-likelihood estimate (MLE) of the fixed effects, random effects, and measurement error using the SAEM algorithm implemented by Monolix software (www.lixoft.eu). For each optimization, the log-likelihood was calculated (log L) and then the corrected Bayesian information criteria (BICc) was computed by Monolix. When models were compared, a model was

considered superior if the BICc increased by more than 4. For all models, time of HIV-1 acquisition was estimated as a key parameter ($t_0$). This initiates the viral dynamics model at a certain time and also allows us to estimate the time of acquisition. The initial conditions were: $V(t_0) = 0.01$ copies/µl, $S(t_0) = \alpha_S/\delta_S$ cells µl⁻¹, and $I(t_0) = V(t_{inf}) \times \gamma/\pi$ cells µl⁻¹. The dynamical system treats viral load in units of copies/µl, so fitting requires scaling $V(t) \times 1000$ to recover the typical viral load per ml. All initial model selection was performed in the placebo population to avoid the effects of VRC01 on model parameters.

We first sought to apply the fully optimized model to the AMP participants. While there is heterogeneity between the study populations, estimating the full model was challenging due to sparse sampling in the AMP study. To address this, we used the fixed effects from our prior modeling of a richer data set[43] (the RV217 trial) model but re-estimated the infection time with random effects and the $\Omega$ matrix parameters comparing three approaches: 1) refitting the random effects standard deviations (diagonals); 2) refitting the correlations (off-diagonals); and 3) refitting the full covariance matrix. Based on comparing BICc, the model refitting the correlations was chosen as the final model for this work with parameters fixed for future PKPD investigation (Supplementary Table 6).

We next investigated the indirect effects of VRC01 viral sensitivity (i.e., viral fitness) on model parameters in the placebo population. Sensitivity was assessed using the threshold (a dichotomous variable) and as a continuous variable on viral parameters: $\kappa, \beta, \pi$ and $n$. As a continuous covariate, IC80 was log10-transformed and centered on the median value $\overline{IC80}$ with following function form on a given parameter: $\log_{10} \theta_k^{pop,adj} = \log_{10} \theta_k^{pop} + b_{\theta_k} \log_{10} \frac{IC80}{\overline{IC80}}$. For the covariate-adjusted model optimization, the $b_{\theta_k}$ and $\omega_k$ terms were fit. All adjusted models showed improved BICc over the unadjusted model with the continuous IC80-adjustment on viral burst size selected as the final adjusted model having the best BICc (Supplementary Table 6). Sensitivity adjustments on more than 1 parameter at once was not tested for parsimony.

## Viral load modeling with PKPD-V

We next augmented the viral load model trained on the placebo recipients to include VRC01 PKPD as follows:

$$\dot{S} = \alpha_S - \delta_S S - \beta(1 - \nu_t) S V$$

$$\dot{I} = \beta(1 - \nu_t) S V - \kappa I^n I \qquad (5)$$

$$\dot{V} = \boldsymbol{\pi} I - \gamma V,$$

where the infection term $\beta(1 - \nu_t) S V$ is modified by the time-varying VRC01 neutralization $\nu_t$. Additionally, based on the placebo model, the modified viral production rate $\boldsymbol{\pi}$ is also explicitly a function of IC80 where higher log10 IC80 (more resistance) linearly reduces lower burst rate, and follows

$$\log_{10} \boldsymbol{\pi} = \log_{10} \pi^{pop} + b_\pi \log_{10} \frac{IC80}{\overline{IC80}} \qquad (6)$$

Where the relationship is centered on the median value (denoted $\overline{IC80}$). Neutralization, ranging from 0 to 1, is expressed using the same function originally used to analyze in vitro titration experiments – i.e., to determine a given virus in vitro IC50 and Hill exponent $h$ (calculated as in Eq. (3) using IC50 and IC80). Here, we assume neutralization instantaneously follows from the sensitivity of the acquired virus against the serum concentration $C_t$ of VRC01 at a given time $t$. However, neutralization is modified by dividing the 50% titer ($PT50 = C_t/IC50$) by the in vivo potency reduction factor $\rho$ to account

for overestimation, leaving

$$\nu_t = \left[1 + \left(PT50/\rho\right)^{-h}\right]^{-1} \qquad (7)$$

We compared four models fit to the VRC01 treatment arms optimizing the $\rho$ parameter: models with and without random effects $\omega_\rho$, and models with and without the indirect IC80 adjustment on burst size. Other parameters were fixed based on the previous modeling. Individual-level PK parameters and acquisition times were determined prior to optimization and fixed as regressors in the model. The best performing model via BICc was the indirect adjusted model without a random effect on $\rho$ (Supplementary Table 6). All viral dynamics model parameters for each individual from the final model can be found in Supplementary Data 3.

The confidence interval for $\rho$ was estimated using standard errors on log10 scale from the model Fisher Information Matrix. The Fisher Information Matrix was estimated using a Markov chain Monte Carlo stochastic approximation algorithm as implemented in Monolix using the Rsmlx package in R.

### Simulations from mathematical model

For each participant, individual-level viral load parameters were selected using the mode of the posterior conditional distribution from the final PKPD viral load model. AMP viral load simulations were then performed using PKPD model incorporating VRC01 PK as determined by the participant's dosing schedule. The AMP viral load trajectories were used to compute viral load summaries as described in the following section. Additionally, participant's viral loads were re-simulated by adjusting the VRC01 dosing, either by removing VRC01 or increasing the doses by several order of magnitudes as a proxy for increasing potency with similar resistance coverage. First positives from the simulations were determined at the matched visit time of the first positive observation from the longitudinal data.

### Viral load summary statistics/metrics

The following viral load outcomes were assessed across treatment arms: first positive, average (mean) viral load, observed set point, model peak, model area under the curve (AUC) of viral load, model set point, viral load upslope, and effective reproductive number (Re). Viral load summaries were always calculated on the log10-transformed scale. Correlations between observed viral load metrics, as well as comparisons of observed viral load metrics across trials and across VRC01 sensitivity groups, are shown in Supplementary Fig. 6.

Model-based summaries were calculated from simulations of viral load trajectories for each participant at the individual-level using the final PKPD model. First positive viral loads corresponded to the first observed HIV-1 RNA PCR above the lower limit of quantification. Average viral load was calculated for each participant as the mean of all observed viral loads prior to ART initiation. The observed set point was calculated as the geometric mean viral load after the observed maximum viral load. If participants had no measurements following peak, they were excluded from the observed set point calculation. Model peak was the maximum viral load from a simulation. Model-predicted AUC for each participant was calculated as the AUC through day 90 after estimated date of acquisition of viral load calculated on one day intervals and normalized by total time. Model-predicted set point was the simulated viral load at day 90 calculated as the mean of values from day 80 to day 90 to smooth any oscillations.

Viral load upslope was determined through regressing simulated viral load on time by fitting a series of models starting at modeled acquisition time with a different end time, ranging from day 3 to day 21. The maximum slope value across the models was used as the viral upslope summary measure.

### Regression model for viral load metrics

Statistical comparisons were performed for first positive, model-estimated set point, and model-estimated viral load AUC using linear regression. For each model, the outcome was regressed on treatment arm, viral sensitivity (dichotomous variable), an interaction between treatment arm and viral sensitivity, and a categorical variable combining protocol and region into 4 total groups (Supplementary Table 1). This implies that covariates were tested for trial, trial arm, geographical region, clade of acquired virus, and viral sensitivity. VRC01 treatment arms were pooled across doses for the analysis. For each model, pairwise comparisons were performed between each combination of treatment arm and viral sensitivity. *P*-values were not adjusted for multiple comparisons unless otherwise stated.

### Reporting summary

Further information on research design is available in the Nature Portfolio Reporting Summary linked to this article.

### Data availability

The data generated in this study are provided in Supplementary Data files 1-3, as well as the public-facing HVTN website (https://atlas.scharp.org/cpas/project/HVTN%20Public%20Data/begin.view?). All individual participant data have been deidentified. The HIV-1 Env clones used in the TZM-bl target cell neutralization assay are available at the GenBank database (https://www.ncbi.nlm.nih.gov/genbank/) under the following accession codes: HVTN 704/HPTN085 sequences, ON980814–ON980967; HVTN 703/HPTN081 sequences, ON890939–ON891092.

### Code availability

All code for data analysis and figure generation is freely available from the authors (https://github.com/FredHutch/AMPVLAnalysis). Python (v 3.10), R (v 4.2), ggplot2 (v 3.4.4)[86], and the Seaborn (v 0.7.1)[87] package were used to streamline the modeling pipeline and make figures.

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

## Acknowledgements

This work was supported by the National Institutes of Health through award numbers R01 AI150500 (E.F. C-O.), UM1 AIO68614 (G.D.T., L.C.), UM1 AIO68635 (Y.H., P.B.G.), 4 R37 AI054165-21 (P.B.G.), UM1 AIO68618, UM1 AIO68619 (M.S.C), UM1 AIO68613, UM1 AIO68617, P30 AIO27757, P30 AIO64518, K25 AI155224 (D.B.R.), the Intramural Research Program of the National Institute of Allergy and Infectious Diseases, and the South African Medical Research Council (SAMRC). The funders had no role in study design, data collection and analysis, decision to publish or preparation of the manuscript. The content is solely the responsibility of the authors and does not necessarily represent the official views of the National Institutes of Health.

## Author contributions

D.B.R., B.T.M., S.T.K., and S.E. conceived the study. D.B.R. and B.T.M. developed the models used in the study. B.T.M. developed the software/code used in the study. D.B.R. and B.T.M. contributed to validation. D.B.R., B.T.M., A.c.dC., Y.H., B.Z., C.A.M., M.J., P.B.G., K.J.B., E.F.C-O, R.R., and P.E. performed analyses. conducted research and/or performed experiments. P.B.G., D.C.M., K.J.B., J.T.S., L.M., N.N.M., C.W., J.I.M., K.E.S., G.D.T., P.A., N.M., J.E.L., M.S.C., L.C., L.N., C.O., P.A.G., M.C., M.E.S., S.T.K., and S.E. provided study materials, laboratory samples,

computing resources or other analysis tools. D.B.R., B.T.M., A.C.dC., C.A.M., and P.B.G. curated data. D.B.R. and B.T.M. performed visualization/data presentation. P.B.G., D.C.M., J.T.S., P.E., C.W., J.I.M., G.D.T., M.S.C., L.C., M.E.S., S.T.K., and S.E. supervised effort on this work. P.B.G., G.D.T., M.S.C., and L.C. acquired funding. D.B.R., B.T.M., and L.N.C. wrote the manuscript. All coauthors reviewed and edited the manuscript and provided approval of the final version.

## Competing interests

The authors declare no competing interests.

## Additional information

[1]Vaccine and Infectious Disease Division, Fred Hutchinson Cancer Center, Seattle, WA, USA. [2]Department of Global Health, University of Washington, Seattle, WA, USA. [3]Public Health Sciences Division, Fred Hutchinson Cancer Center, Seattle, WA, USA. [4]Department of Biostatistics, University of Washington, Seattle, WA, USA. [5]Department of Surgery, Duke University Medical Center, Durham, NC, USA. [6]Perelman School of Medicine, University of Pennsylvania, Philadelphia, PA, USA. [7]Department of Medicine, University of Washington, Seattle, WA, USA. [8]National Institute for Communicable Diseases, National Health Laboratory Service, Johannesburg, South Africa. [9]Antibody Immunity Research Unit, Faculty of Health Sciences, University of the Witwatersrand, Johannesburg, South Africa. [10]Centre for the AIDS Programme of Research in South Africa, University of KwaZulu-Natal, Durban, South Africa. [11]Division of Medical Virology, Faculty of Health Sciences, University of Cape Town and National Health Laboratory Service, Cape Town, South Africa. [12]Department of Microbiology, University of Washington, Seattle, WA, USA. [13]Center for Human Systems Immunology, Duke University, Durham, NC, USA. [14]Departments of Surgery, Immunology, and Molecular Genetics and Microbiology, Duke University, Durham, NC, USA. [15]Family Health International, Durham, NC, USA. [16]Clinical Trials Research Centre, University of Zimbabwe College of Health Sciences, Harare, Zimbabwe. [17]Vaccine Research Center, National Institute of Allergy and Infectious Diseases, National Institutes of Health, Bethesda, MD, USA. [18]Institute for Global Health and Infectious Diseases, The University of North Carolina at Chapel Hill, Chapel Hill, NC, USA. [19]Department of Laboratory Medicine and Pathology, University of Washington, Seattle, WA, USA. [20]South African Medical Research Council, HPRU, Durban, South Africa. [21]Desmond Tutu HIV Centre, Institute of Infectious Disease and Molecular Medicine and Department of Medicine, University of Cape Town, Cape Town, South Africa. [22]Division of Infectious Diseases, Department of Medicine, University of Alabama at Birmingham, Birmingham, AL, USA. [23]Facultad de Medicina Humana, Universidad Nacional de la Amazonia Peru, Iquitos, Peru. [24]Division of Infectious Diseases, Department of Medicine, Vagelos College of Physicians and Surgeons, New York-Presbyterian/Columbia University Irving Medical Center, New York, NY, USA. [25]GreenLight Biosciences, Medford, MA, USA. [26]Division of Infectious Diseases, Department of Medicine, Emory University School of Medicine, Atlanta, GA, USA. [27]These authors contributed equally: Daniel B. Reeves, Bryan T. Mayer. [28]These authors jointly supervised this work: Shelly T. Karuna, Srilatha Edupuganti. ✉e-mail: dreeves@fredhutch.org

