## [Peer Review File · Nature Communications]

High monoclonal neutralization titers reduced breakthrough HIV-1 viral loads in the Antibody Mediated Prevention trialsEditorial Note: This manuscript has been previously reviewed at another journal that is not operating a transparent peer review scheme. This document only contains reviewer comments and rebuttal letters for versions considered at Nature Communications.

Reviewers' Comments:

Reviewer #1:

Remarks to the Author:

In the revised manuscript, the authors have addressed my comments satisfactorily.

Reviewer #2:

Remarks to the Author:

The authors have done a nice job of addressing my comments. I have only a minor comment now. In the Discussion, lines 405-407, the authors mention that the in vivo potency of VRC01 'required to suppress' viremia was 600-fold lower than would be expected from serum concentrations and in vitro IC80s. This gives the impression that the potency of VRC01 in vivo is higher than in vitro, which is not the case. I suggest rewording the sentence to the in vivo potency that 'explains the suppression' of viremia observed is 600 times lower than...

Reviewer #3:

Remarks to the Author:

This reviewer thanks the authors for carefully addressing all concerns and adding additional analyses that have strengthened this manuscript.

Reviewer #1 (Remarks to the Author):

In the revised manuscript, the authors have addressed my comments satisfactorily.

Response: We are glad to hear this.

Reviewer #2 (Remarks to the Author):

The authors have done a nice job of addressing my comments. I have only a minor comment now. In the Discussion, lines 405-407, the authors mention that the in vivo potency of VRC01 'required to suppress' viremia was 600-fold lower than would be expected from serum concentrations and in vitro IC80s. This gives the impression that the potency of VRC01 in vivo is higher than in vitro, which is not the case. I suggest rewording the sentence to the in vivo potency that 'explains the suppression' of viremia observed is 600 times lower than...

Response: Thank you for the suggestion, we agree that we want to avoid any suggestion that VRC01 is more potent in vivo than in vitro. We have revised the sentence to

“Here, we estimated that the in vivo VRC01 PT80 titer required to suppress viremia was 600-fold higher than would be expected from serum concentrations and in vitro IC80 titers, suggestive of in vivo potency reduction.

There are several possible explanations for in vivo potency reduction....”

Reviewer #3 (Remarks to the Author):

This reviewer thanks the authors for carefully addressing all concerns and adding additional analyses that have strengthened this manuscript.

Response: Thanks for the positive assessment.